# Laser Ablation Inductively Coupled Plasma Spectrometry: Metal Imaging in Experimental and Clinical Wilson Disease

**Sabine Weiskirchen, Philipp Kim and Ralf Weiskirchen ***

Institute of Molecular Pathobiochemistry, Experimental Gene Therapy and Clinical Chemistry, RWTH University Hospital Aachen, D-52074 Aachen, Germany; sweiskirchen@ukaachen.de (S.W.); hkim@ukaachen.de (P.K.)

**\*** Correspondence: rweiskirchen@ukaachen.de; Tel.: +49-(0)241-8088683

**Abstract:** Wilson disease is an inherited disorder caused by mutations in the *ATP7B* gene resulting in copper metabolism disturbances. As a consequence, copper accumulates in different organs with most common presentation in liver and brain. Chelating agents that nonspecifically chelate copper, and promote its urinary excretion, or zinc salts interfering with the absorption of copper from the gastrointestinal tract, are current medications. Also gene therapy, restoring *ATP7B* gene function or trials with bis-choline tetrathiomolybdate (WTX101) removing excess copper from intracellular hepatic copper stores and increasing biliary copper excretion, is promising in reducing body's copper content. Therapy efficacy is mostly evaluated by testing for evidence of liver disease and neurological symptoms, hepatic synthetic functions, indices of copper metabolisms, urinary copper excretions, or direct copper measurements. However, several studies conducted in patients or Wilson disease models have shown that not only the absolute concentration of copper, but also its spatial distribution within the diseased tissue is relevant for disease severity and outcome. Here we discuss laser ablation inductively coupled plasma spectrometry imaging as a novel method for accurate determination of trace element concentrations with high diagnostic sensitivity, spatial resolution, specificity, and quantification ability in experimental and clinical Wilson disease specimens.

**Keywords:** metal; imaging; genetic disease; *ATP7B*; copper; diagnostic; animal model; therapy; neuropsychiatric symptoms; liver disease

## 1. Introduction

Wilson disease (WD) is a rare, inherited autosomal recessive copper (Cu) overload disease, in which excess Cu accumulates in the liver, brain, and other tissues. It is caused by mutations within the *ATP7B* gene encoding a plasma membrane Cu-transport protein [1]. It is assumed that the disease has a general prevalence of ~1:30,000. However, the frequency can be significantly higher (up to 1:1130) in populations with a low rate of immigration [2,3]. Excessive free Cu is rather toxic, causing a broad range of clinical manifestations, including liver cancer and severe psychiatric and neurological symptoms [4]. The symptoms in WD patients vary widely and usually appear between the ages of six years and 20 years, but there are also cases in which the disease manifests in advanced age [5]. In addition, there is growing evidence that modifier genes and epigenetic mechanisms significantly affect the phenotypic presentation and contribute to the pathogenesis of WD [6]. Accumulation and toxicity of Cu is progressive and ultimately fatal without specific therapy. Therefore, once a diagnosis of WD is established, clinical practice guidelines for WD recommend long-term or even lifelong treatment with appropriate metal chelators (D-penicillamine, trientine) enhancing urinary excretion of Cu [4,7]. In addition to chelating agents, the blockade of Cu by zinc (Zn) salts and the avoidance of intake of foods

and water with high Cu concentrations is recommended to remove excess Cu from the body [4,7,8]. The main indicators in treatment monitoring of Cu balance after chelating therapy are the daily urinary Cu excretion and the normalization of non-ceruloplasmin-bound Cu [8]. However, all present pharmacological WD treatments are associated with several limitations, including the persistence of neurological symptoms, the occurrence of drug-related adverse effects, and a potential poor compliance of patients resulting from complex and cumbersome treatment regimens [9]. Therefore, more effective drugs or the establishment of more effective treatment modalities in WD are urgently needed.

In recent years, several preclinical animal models of WD were established [10–12]. These models are highly helpful to test novel drugs, examine Cu distribution and metabolism, and to improve the understanding of mechanisms by which Cu exerts its toxic effects [10]. In particular, these experimental models with identical genetic background are highly suitable to analyze aspects of Cu homeostasis and overload without the high variability observed in humans.

In this regard, the combination of respective animals models with novel quantitative biometal imaging techniques such as (synchrotron) X-ray fluorescence microscopy (XFM), secondary ion mass spectrometry (SIMS), particle induced X-ray emission (PIXE), energy dispersive X-ray spectroscopy (EDS), and laser ablation inductively coupled plasma mass spectrometry (LA-ICP-MS) that allow to speciate and locate a number of elements with high accuracy, reproducibility, and expenditure within the tissue, have significantly contributed to the knowledge in the detection and quantification of metal-associated diseases [13,14].

In the present review, we discuss how LA-ICP-MS imaging (LA-ICP-MSI) contributed to the knowledge in the pathogenesis of WD. Moreover, we highlight the advances in using LA-ICP-MSI techniques in experimental and clinical studies to monitor disease progression, detect therapeutic effects of drugs, or to document the reversal of Cu overload by gene therapeutic approach.

## 2. Copper Homeostasis and Overload

An adult usually takes up ~2 mg of Cu each day through food and beverages, while the daily requirement is estimated to be approximately 0.8 mg per day [15]. To avoid Cu poisoning, Cu homeostasis in healthy subjects is tightly controlled (Figure 1). Cu is acquired from the diet by intestinal absorption and is distributed throughout the body [16]. In the body, Cu is mainly transported to the liver and incorporated into ceruloplasmin or excreted into the bile. In general, higher Cu in the diet is associated with higher Cu absorption, although the correlation is nonlinear. Under conditions of Cu excess, extra Cu can be sequestered, safely stored, and presumably accessed later when Cu levels decrease [17]. However, a great fraction of excess Cu is eliminated in feces, both as absorbed an unabsorbed metal ions and from biliary excretion, averaging 0.5–1.3 mg per day [18]. Furthermore, during phases of excess Cu, metallothionein is induced to mask Cu toxicity by binding tightly to Cu ions render them redox inactive [19]. Although the body of a healthy person contains approximately 100 mg Cu in total, only small fractions of free Cu are found in the serum and urine [15]. However, serum and urinary levels of Cu in patients suffering from liver disease such as chronic hepatitis B infection can be significantly higher [20].

In patients suffering from WD, Cu distribution within the body completely changes. While Cu absorption is normal, there is no incorporation of Cu into ceruloplasmin and no excretion into the bile. This provokes a significant increase of Cu in hepatocytes, elevated concentrations of free Cu plasma, and increased quantities of urinary Cu [21]. The elevated concentrations of free Cu are toxic and are associated with mitochondrial dysfunction, oxidative stress, cell membrane damage, enzyme inhibition, and formation of DNA cross-links [1]. In the liver, this predisposes for liver cirrhosis and hepatocellular carcinoma. In the brain, Cu overload induces dysfunction of the blood–brain barrier and prominent demyelination of the central nervous system, providing the basis for the occurrence of many neurological symptoms and disorders of the extrapyramidal motor system.

Clinically, one of the most striking clinical signs appearing in many WD patients is the occurrence of corneal Kayser–Fleischer rings consisting of annular Cu deposits in the Descemet's membrane [5].

Other hallmarks are low serum ceruloplasmin concentrations (<20 mg/dL) and increased hepatic Cu concentrations (>250 µg/g dry weight) [4]. Interestingly, a large number of WD patients with exclusive hepatic forms do not develop visible Kayser–Fleischer rings or reduced ceruloplasmin levels. In these patients, the appearance of elevated Cu in the liver alone is considered as the best, but not exclusive, laboratory WD indicator [22].

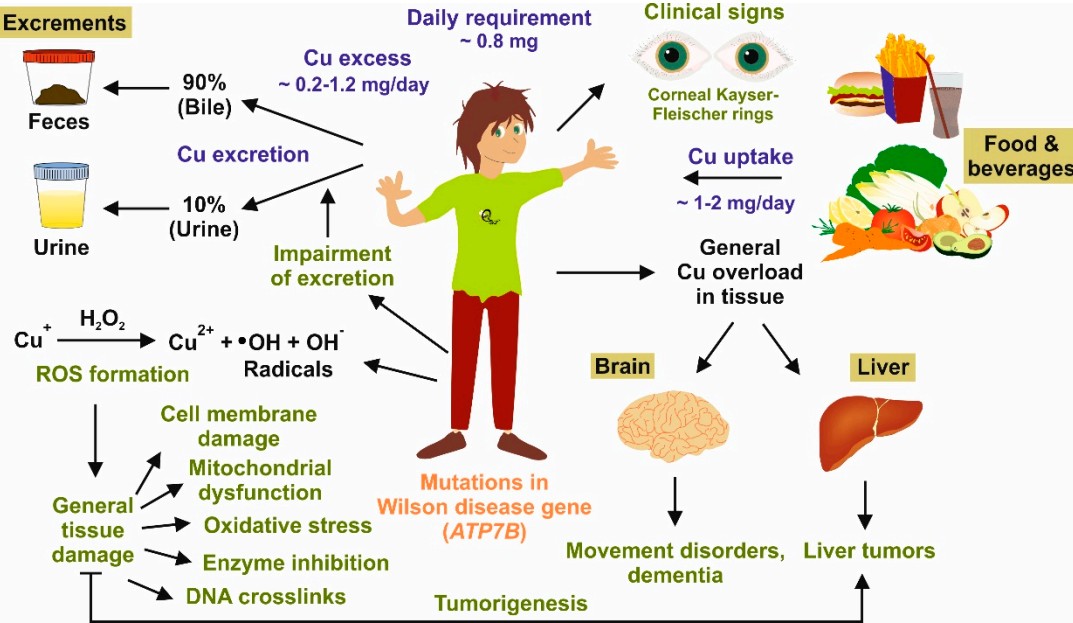

**Figure 1.** Body copper homeostasis and clinical manifestations in Wilson disease. Adults typically require ~0.8 mg Cu/day, while ingesting ~1.0–2.0 mg/day through food and beverages. The body excretes ~90% of excess Cu through the bile/feces and ~10% through the kidneys. In Wilson disease patients, the *ATP7B* gene is mutated resulting in Cu overload, inevitably leading to progressive liver and neurological dysfunction. Furthermore, Cu accumulation is associated with the generation of reactive oxygen species (ROS), mitochondria dysfunction, enzyme inhibition, cell membrane damage, and DNA modification. Clinically, Cu overload manifests in the occurrence of corneal Kayser–Fleischer rings, liver tumors, highly variable neurologic peculiarities, psychiatric abnormalities reflecting in movement disorders, and dementia.

In sum, diagnosis and classification of Cu overload occurring in WD, and its discrimination from other related metal metabolism or storage diseases, requires a high index of suspicion and is based on a combination of clinical signs, biochemical tests, measurements on hepatic Cu, and mutation analysis [21,23].

## 3. Copper Distribution in Wilson Disease Tissue

As discussed, the determination of hepatic Cu concentration is important in the diagnosis of WD. However, independent studies have demonstrated that hepatic Cu for example is unevenly distributed in WD in the cirrhotic stage. This was first documented for example in a study, in which Cu distribution was studied in the cirrhotic liver of a patient who died of WD [24]. In this study, the authors measured Cu content in 38 samples prepared from a liver slice extending from the left to the right lobe. While the hepatic Cu concentration had a mean of 1370 µg/g dry tissue, the Cu concentration as determined by induced coupled plasma atomic emission spectroscopy and histochemical analysis showed a striking variability up to 2–3 fold (880–2100 µg/g dry tissue), with significant differences even between adjacent samples. Moreover, highest quantities of Cu were found in periportal hepatocytes, while areas of parenchyma negative for Cu could be detected in close proximity of regions full of Cu granules [24]. Therefore, it is conceivable that the irregular distribution of Cu within the liver can

result in underestimation of hepatic Cu. This may occur, in particular, in later stages of WD or during monitoring of Cu chelating therapy [4]. Furthermore, uneven distribution of Cu is also found at the subcellular level. A historical landmark paper localized a significant portion of hepatic Cu excess to the mitochondrial fraction in WD patients [25]. Interestingly, in WD patients a proportion accounting for 35% of the total tissue Cu were found in the mitochondria, a value which is ~40 times higher than those observed in livers of healthy adults [25].

All these findings clearly indicate that not only the absolute Cu concentration, but also the spatial distribution within the affected tissues/cells is diagnostically relevant. In this regard, metal bioimaging by LA-ICP-MSI allowing measurements with spatial resolution has significant diagnostic advantages in comparison to other traditional or routine methods presently used for Cu quantitation. This methodology not only provides information about mean concentrations of a specific metal, but is suitable to simultaneously measure absolute element quantities of a large number of metals and metalloids with a high spatial resolution (see below). Therefore, it is much more favorable than simple histochemical detection of excess Cu by various stains, including rhodanine, rubeanic acid, Timm silver stain, orcein stain, or others. Like all other histochemical staining procedures, the rhodanine stain, for example, is suitable to document widespread and heavy deposition of Cu granules throughout the cirrhotic nodule (Figure 2), but is not appropriate to determine absolute Cu concentrations within the overloaded tissue. Moreover, other reports showed that in some cases histochemical Cu stains resulted in noninterpretable diffuse and faint staining pattern or failed completely [26,27]. So again, more powerful analytical methods such as LA-ICP-MSI ensuring highly sensitive and quantitative element imaging are more favorable than the mentioned histochemical stains.

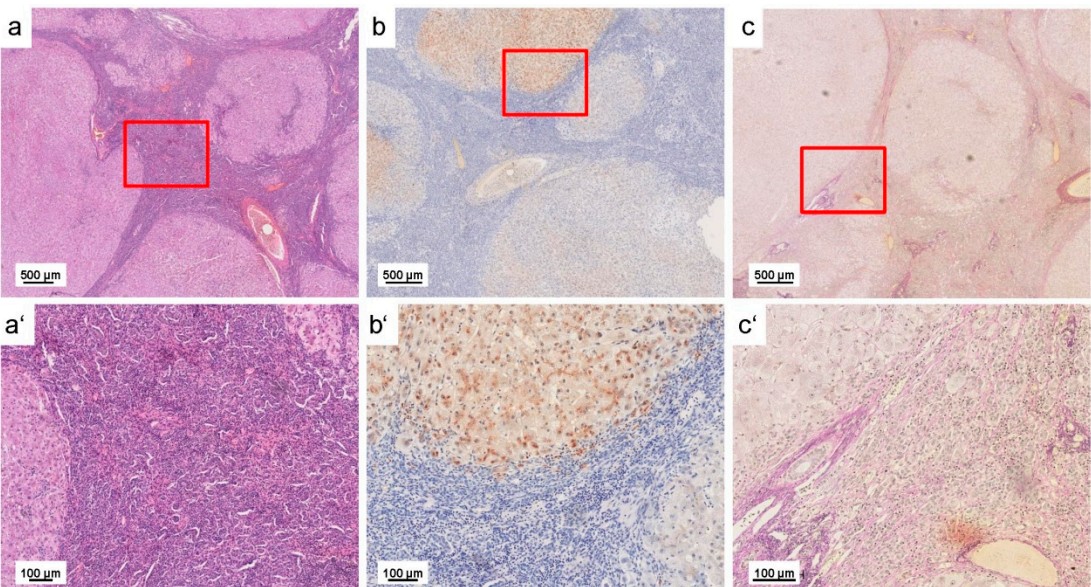

**Figure 2.** Liver pathology in Wilson disease as assessed by standard histochemical stains relying on chemical reactions. Liver specimens from a patient suffering from Wilson disease were stained with (**a** and **a'**) hematoxylin & eosin, (**b** and **b'**) rhodanine, and (**c** and **c'**) Verhoeff-van Gieson. The hematoxylin & eosin and the Verhoeff-van Gieson stains show established bridging fibrosis (cirrhosis) and liver damage, respectively. Cytoplasmic Cu accumulation is well-documented in the rhodanine stain. Space bars represent 500 μm (**a**, **b**, and **c**) or 100 μm (**a'**, **b'**, and **c'**), respectively.

## 4. Laser Ablation Inductively Coupled Plasma Spectrometry Imaging

LA-ICP-MSI can be used for a large number of applications. In brief, the methodology uses a finely-focused laser beam with micrometer spot size to ablate a biological sample by line-by-line scanning along the sample (Figure 3). Thereafter, the ablated material is transported into the inductively coupled plasma (ICP) source of the mass spectrometer using an inert carrier gas such as argon. Here,

the material is vaporized, atomized, and ionized. The charged ions are then directed into a mass spectrometer (MS) and separated according to their mass to charge (*m/z*) ratio. LA-ICP-MSI has multielemental and isotopic selectivity and allows the detection of different elements (metals and non-metals) with spatial resolution of 5–150 μm at concentrations as low as one part per quadrillion ($10^{15}$) when the analyzed material is free of interfered background isotopes [13,14]. In addition, LA-ICP-MSI provides simple spectra, isotopic information about a large variety of elements, and has a rapid sample throughput. Compared to other mass spectrometry techniques, it allows direct analysis of 5–100 μm thick tissue cryosections without any time-consuming sample preparation procedure or the need for matrix deposition on the tissue surface as required by the widely used matrix-assisted laser desorption/ionization mass spectrometry (MALDI-MS).

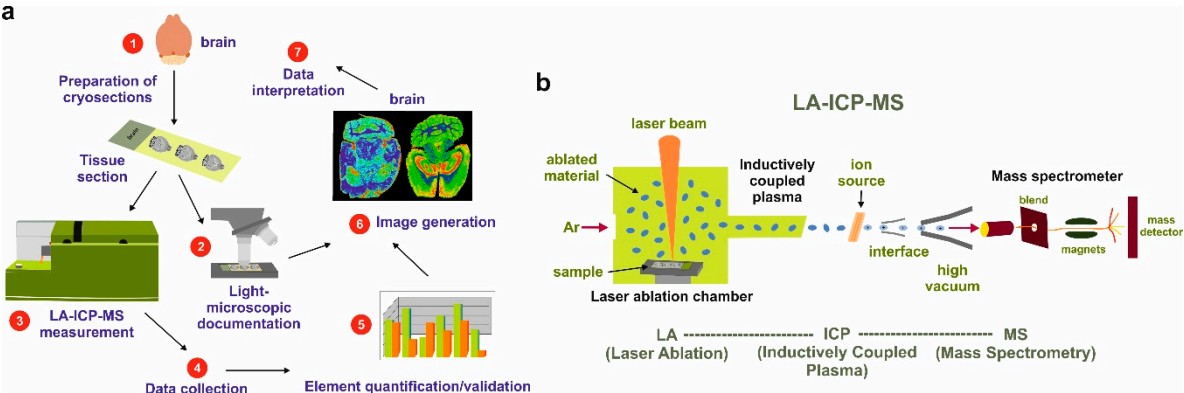

**Figure 3.** Simplified laser ablation inductively coupled plasma mass spectrometry imaging (LA-ICP-MSI) workflow. (**a**) LA-ICP-MSI of a tissue sample requires seven sequential steps, including (1) the preparation of a cryosection from a specimen, (2) light-microscopic documentation of the section, (3) ablation of biological material and element measurement, (4) data collection, (5) calculation of relative or absolute element concentrations, (6) production of elements maps, and finally (7) interpretation of the LA-ICP-MSI results. (**b**) For the LA-ICP-MSI measurement, the sample is positioned in the laser ablation (LA) chamber containing a mobile sample tray enabling line-by-line scans in all directions. Simplified, a focused laser beam ablates biological material, which is transported in an inert carrier gas stream into the inductively coupled plasma (ICP). There the sample is decomposed into its constituent elements and transformed into ions by electromagnetic induction. The ions are subsequently transferred into a high vacuum mass spectrometry (MS) analysis chamber and identified according to their mass/charge (*m/z*) ratio. The concentration of each element is directly correlated to the signal intensities.

This method further allows the ablation of nonconducting biological sample surfaces without charging effects, which usually occur during secondary ion mass spectrometry SIMS measurements. However, one disadvantage of LA-ICP-MSI is represented by its destructive action to the specimen. After laser bombardment of the sample, a small pit encompassing the size of the analytical volume is left behind. This results in a typical dashed appearance of the probe, which reflects the line-by-line scan of the laser during the analysis (Figure 4).

However, it should be critically mentioned that in some cases it can be difficult to quantify individual metals in a sample. This is either due to a lack of standard reference materials, which often requires establishing homemade matrix-matched standards for quantification [28], or to isobaric interferences of different elements whose isotopes share a common mass and create spectral interferences, thereby preventing detailed chemical resolution [29]. Another potential hindrance in the usage of LA-ICP-MSI in daily routine is the final processing of measured data that can be very elaborate. In most cases, LA-ICP-MSI generates raw data files requiring extensive editing and formatting before data can be used to generate high-quality elemental distribution maps. For this purpose, a large abundance of commercial, open-source, or in-house programs and software routines have been introduced [30]. The plethora of software available used for LA-ICP-MSI data mining is multifarious allowing depicting

results in many formats. However, this large abundance might prevent the exchange of data between different laboratories [30].

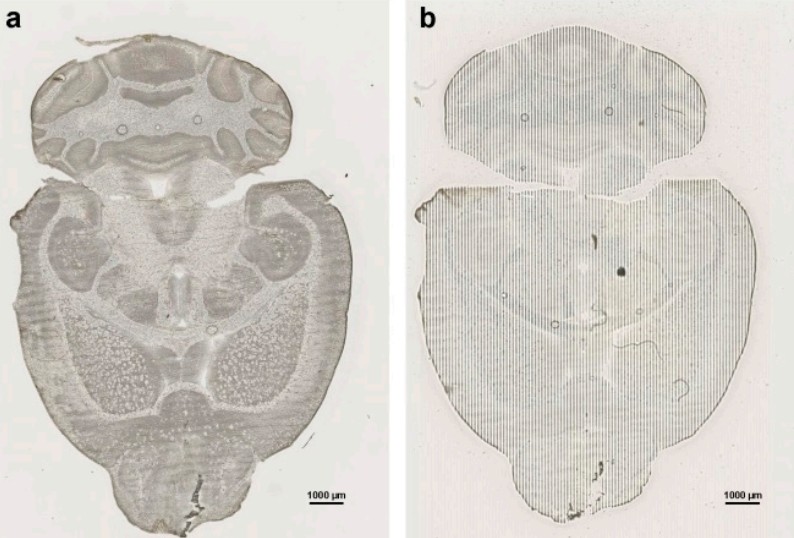

**Figure 4.** Macroscopic appearance of serial cuts taken from a mouse brain before (**a**) and after (**b**) laser ablation. Please note the typical lines in the ablated sample resulting from the laser bombardment performed as a line-by-line scan with a laser spot size of 80 μm of the analyzed sample. Size bars represent 1000 μm.

Moreover, in all these studies it is essential that identical brain regions are compared. In the workflow of our laboratory, the prepared section is first scanned in a digital slide scanner for documentation before it is analyzed by LA-ICP-MSI. After the measurement, both the LA-ICP-MSI and the bright field images are superimposed to accurately match the measured metal concentrations to the correct brain region. To guarantee that we compare sections taken from the identical brain regions, we prepare consecutive sections of each brain and compare identical ones by referring to appropriate anatomical references atlases such as the Allen Reference Atlases [31].

## 5. Wilson Disease Studies Using LA-ICP-MSI

Within the brain, metals have a highly compartmentalized distribution [32,33]. Therefore, imaging the spatial distributions of metals in brain samples has provided unique insight into the biochemical architecture of the brain, allowing direct correlation of neuroanatomical regions and metal-dependent processes as well as changes in metal homeostasis and formation of disease such as Alzheimer's disease, Parkinsonism, Amyotrophic Lateral Sclerosis, and other neurodegenerative disorders [32–36]. In regard to Wilson disease, a number of reports using LA-ICP-MSI for element bioimaging in experimental or clinical Wilson disease are available (Table 1).

**Table 1.** Studies using LA-ICP-MSI in experimental or clinical Wilson disease research.

| Specimens | Major Findings | Refs |
|---|---|---|
| Liver sections of *Atp7b*$^{-/-}$ and wild type mouse (each *n* = 5) | The average hepatic concentrations of Cu, Mn, Fe, and Zn were 4, 0.7, 41, and 18 µg/g tissue in control samples, while they were 143, 0.6, 80, and 32 µg/g tissue in livers of *Atp7b*$^{-/-}$ mouse | [37] |
| Brain sections of wild type (*n* = 8) and *Atp7b*$^{-/-}$ (*n* = 9) mice in the age range of 11-24 months | Cu was increased proportionally during ageing throughout all cerebral regions; *Atp7b* null mice showed ~2-fold stable increase of Cu throughout brain parenchyma; Cu was ~3.5-fold decreased in periventricular regions | [38] |
| Liver sections of wild type and *Atp7b*$^{-/-}$ mice and human WD liver samples | Liver sections of 10 months old *Atp7b* null mice and patients with WD showed irregular (patchy) Cu distribution with high regional concentrations; age-dependent accumulation of hepatic Cu was strictly associated with a simultaneous increase in iron (Fe) and Zn; human liver samples confirmed accumulation of hepatic Fe and Zn in WD patients; tumorigenic regions are highly enriched in Cu | [39] |
| Paraffin-embedded human liver needle biopsy (*n* = 2) | WD patients showed inhomogeneous Cu distribution and high Cu concentrations of up to 1200 µg/g; inverse correlation of regions with elevated Cu and region with high Fe concentrations | [40] |
| Liver of D-Penicillamine (DPA)-treated *PINA/Atp7b*$^{-/-}$ (LPP$^{-/-}$)** rats and a liver samples from a patient before and after DPA treatment | DPA-treatment resulted in a significant decrease in hepatic Cu by more than a factor two; Cu distribution maps after DPA treatment were highly inhomogeneous and lowest Cu concentrations were found in direct proximity to blood vessels | [41] |
| Human stained and unstained liver needle biopsies (*n* = 8) | When comparing unstained and rhodanine-stained sections of each WD liver sample, unstained sections showed distinct structures of Cu distribution, while rhodanine-stained sections revealed blurred Cu distribution with 20–90% decreased concentrations | [42] |
| Liver sections of untreated or AAV-AAT-*co*-miATP7B-treated *Atp7b*$^{-/-}$ mice (*n* = 5) | While the mean of hepatic Cu was 112.7 ± 13.3 µg/g liver tissue in the untreated group, the delivery of the transgene reduced Cu content to a mean of 43.3 ± 3.6 µg/g liver tissue; removal of Cu provoked a simultaneous decrease in hepatic Fe (314 ± 38 vs. 150.2 ± 25.2 µg/g liver tissue) and a slight reduction in hepatic Zn (43.1 ± 3.5 vs. 32.4 ± 4.3 µg/g liver tissue) | [43] |
| Brain sections of untreated (*n* = 5) or AAV-AAT-*co*-miATP7B*-treated *Atp7b*$^{-/-}$ mice (*n* = 6) | Brains of animals receiving the transgene had overall lower concentrations of total cerebral Cu (3.8 ± 0.2 vs. 3.05 ± 0.17 µg/g brain tissue), most prominently noticeable in the cerebellum, cerebellar white matter, corpus callosum, 3$^{rd}$ and 4$^{th}$ ventricles, and surrounding tissue, and a slight decrease in the basal ganglia; the content in the *Atp7b*$^{+/-}$ control mice that showed no alterations in Cu metabolism was 2.34 ± 0.09 µg/g. Concentrations of Fe, Zn, Mn, Na, Mg, K, Ca, P, Cr, Ni, and Pb were unaffected by the therapeutic approach | [44] |
| Liver samples from *PINA/Atp7b*$^{-/-}$ (LPP$^{-/-}$) rats treated with Methanobactin OB3b*** | Hepatic Cu hotspots were effectively removed by treatment with Methanobactin OB3b; Cu re-accumulation was observed after interruption of therapy | [45] |

* The respective transgene is based on an adeno-associated virus (AAV) serotype 8 encoding a human *ATP7B* mini (mi) gene codon-optimized (*co*) under transcriptional control of the liver-specific α1-antitrypsin (AAT) promoter. ** This rat strain carries a deletion in *Atp7b* and produces an alternative *Atp7b* gene product termed Pineal night-specific ATPase (PINA) [46]. *** This compound is a commonly studied methanobactin composed of 9 amino acid residues (i.e., Leu-Cys-Gly-Ser-Cys-Tyr-Pro-Cys-Ser-Cys-Met) with two oxazolone rings, which take part in binding to Cu ions.

In a first study, liver sections from *Atp7b*$^{-/-}$ mice were analyzed by LA-ICP-MSI [37]. This mouse model features properties of human WD during aging [47]. In particular, the gene defect leads to (i) an accumulation of Cu in the liver, kidneys, and brain to a level up to 60-fold greater than normal by 5 months of age; (ii) slight neurologic abnormalities in some animals; (iii) increased urinary Cu excretion; (iv) increased metallothionein expression; and (v) alterations in lipid metabolism and cell cycle machinery [47,48] (Figure 5). Moreover, the elevated intracellular and nuclear Cu deposits in the hepatocytes induce a severe liver pathology, including steatosis, inflammation, dysplasia and ultrastructural changes, bile duct proliferation, fibrogenesis, and neoplastic proliferation [47].

When comparatively analyzing trace metals in liver sections taken from wild type and *Atp7b* null mice, the elements Cu, Fe, and Zn showed a significant higher concentration in the diseased animals [37], while the average concentration of hepatic Mn was the same in control and the experimental WD model (Figure 6). This finding demonstrates that *Atp7b* mutation not only provokes hepatic Cu overload, but is further associated with critical alterations of the hepatic metallome [37].

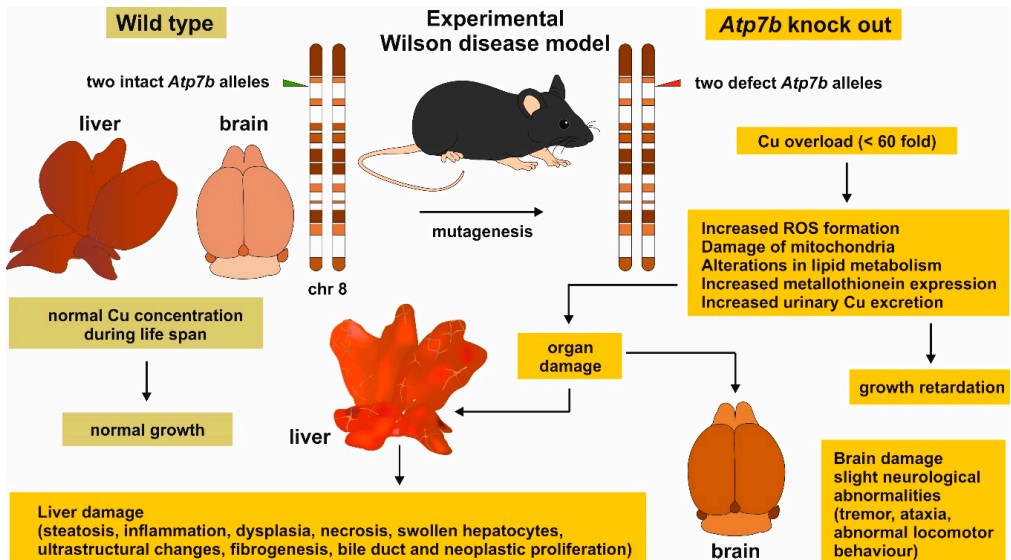

**Figure 5.** *Atp7b^{−/−}* mice as a useful model for studies on Wilson disease. In mice, the genetic disruption of both *Atp7b* alleles located on chromosome 8 results in a phenotype resembling human Wilson disease. Respective mice display a gradual accumulation of Cu in liver, brain, and other organs. This provokes increased formation of reactive oxygen species (ROS), mitochondrial damage, measurable alterations in lipid metabolism, elevated metallothioneine expression, and increased urinary Cu excretion. As a consequence, Cu overload provokes liver and brain damage. In contrast, in wild type mice, Cu concentration is more or less stable in the complete life span.

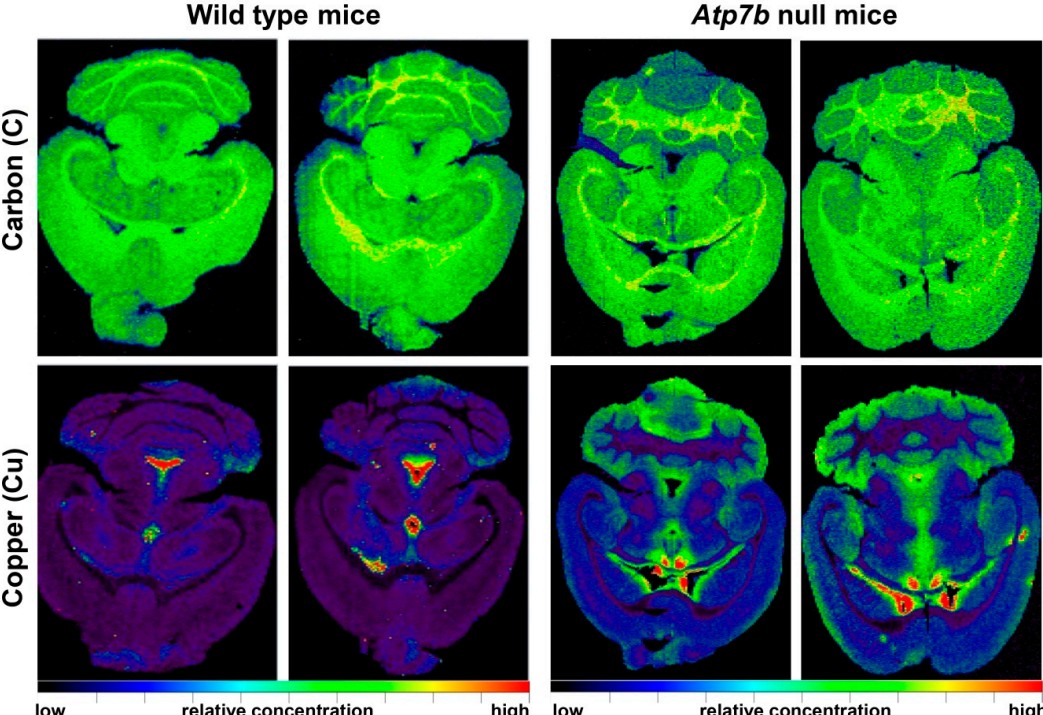

**Figure 6.** Comparative LA-ICP-MSI of Cu in brains of wild type and *Atp7b* null mice. Two representative brain sections, each taken from adult wild type (left) and *Atp7b* null mice (right) were imaged for carbon (C) and copper (Cu). Depicted are relative concentrations of Cu which were normalized to the content of C. The increase of cerebral Cu in *Atp7b* null mice predominantly affects specialized brain areas. Details about age-dependent accumulation of Cu in *Atp7b* null mice are given elsewhere [37,38].

In a subsequent study, this exploratory $Atp7b^{-/-}$ mouse model was employed to analyze the age-dependent accumulation of Cu in hepatic tissue [39]. The respective LA-ICP-MSI study revealed that in *Atp7b* null mice, Cu irregularly accumulates in the liver during aging with extremely high quantities in regions with tumorigenic nodules, suggesting a direct correlation of Cu concentration, tissue damage, and tumor formation. In contrast, hepatic Cu concentration stayed constantly low in wild type control mice. The accumulation of hepatic Cu was highest in animals at age 14 months and decreased thereafter, possibly reflecting ongoing parenchymal damage of the liver [39]. In agreement with the former study, a simultaneous accumulation of Zn and Fe in the diseased liver tissue was found [39]. Interestingly, also the measurement in human WD liver samples showed the same concurrent accumulation of Cu, Zn, and Fe.

When analyzing brain sections of $Atp7b^{-/-}$ mice, an ~2-fold increase of Cu throughout the brain parenchyma combined with reduced quantities in periventricular regions (PVR) was found, corroborating the view of the PVRs as efflux compartments with active transport of Cu into the cerebrospinal fluid [38]. Compared to wild type animals, Zn was found to increase in cerebral regions enriched in Cu, but not in regions that typically show elevated concentrations of this trace element. However, in the brains of *Atp7b* null mice, the quantities of Fe and Mn were unchanged in all brain areas [38].

In another study, elemental bioimaging of Cu and Fe by means of LA-ICP-MSI was performed in paraffin-embedded liver needle biopsy specimens from two WD patients and a control sample originating from a patient suffering from alcoholic liver cirrhosis [40]. The elemental distribution maps for Cu and Fe were highly inhomogeneous in both WD samples, showing highly variable concentrations within the tissue ranging from 0 to 1000 µg/g tissue for Cu and 0 to 2500 µg/g tissue for Fe, respectively. Interestingly, the concentrations of both trace elements were inversely correlated to each other [40]. In contrast, the biopsy sample taken from the alcoholic showed homogenous Cu distribution within the tissue with overall low concentrations ranging from 0 to 50 µg/g liver tissue, while Fe was irregularly distributed within the diseased tissue (0–2500 µg/g tissue) [39], possibly reflecting the typical decreasing gradients from periportal to centrilobular areas of the liver visible in LA-ICP-MSI [39].

The suitability of LA-ICP-MSI for visualization of element distributions in livers of clinical and experimental samples was also documented in a proof-of-concept study, in which therapeutic effects of D-penicillamine (DPA) were investigated [41]. While liver samples obtained from two untreated LPP null rats lacking functional *Atp7b* genes showed high hepatic Cu concentrations (402 ± 156 or 267 ±131 µg/g tissue), the treatment with DPA for 36 or 37 days resulted in a significant decrease in Cu (121 ± 35 or 192 ±101 µg/g tissue). However, the Cu distribution in respective animals was highly inhomogeneous, with the lowest concentrations in direct proximity to blood vessels. In addition, the decrease in Cu corroborated with a reduction in Fe by a factor of 2 to 3, while the concentration of Zn was not altered [41]. In the same study, the total elemental concentrations of Cu, Fe and Zn in a human liver sample taken from patients that received DPA was determined to 34, 83, and 89 µg/g tissue, respectively. The distribution of hepatic Cu and Fe in this patient was inversely correlated, suggesting that the chelating therapy potentially replaces Cu by Fe when the drug is applied for prolonged times [41].

A diagnostically highly relevant paper investigated the influence of rhodanine and hematoxylin and eosin (H&E) staining on the Cu distribution and concentration in WD liver needle biopsy samples by means of LA-ICP-MSI [42]. In this study, unstained and rhodanine-stained parallel sections of eight WD liver samples were analyzed. Importantly, the unstained sections showed distinct Cu distribution with high regional Cu concentrations, while sections measured after rhodanine staining Cu showed 20 to more than 90% lower Cu values that had an overall blurry distribution, while the standard H&E stain had no impact on the Cu distribution or concentrations [42]. Based on these findings, the authors correctly concluded that the rhodanine staining procedure leads to a removal of Cu and a potential underestimation of the actual concentration, and further combination of H&E stain with subsequent

analysis by means of LA-ICP-MSI is a diagnostic option to gain additional information on the elemental distribution within a pathology specimen [42].

Other studies used LA-ICP-MSI to evaluate the therapeutic efficacy of drugs in treatment of experimental WD. In the above mentioned LPP *Atp7b*$^{-/-}$ rats, the therapeutic treatment with Methanobactin OB3b (MB), for either five weeks at doses of 62.5 mg/kg bodyweight three times per week or alternatively therapy with twice daily doses of MB for eight consecutive days at a dose of 150 mg/kg bodyweight, resulted in reduced Cu content less than half of those of untreated rats [45]. Although untreated animals showed higher Fe and Zn concentrations, the treatment with MB had no significant impact on these elements. In agreement with a previous study [39], the authors could demonstrate that Cu is highly irregularly distributed in the overloaded tissue, forming large, but distinct hotspots in which concentrations of up to 890 µg/g tissue can be found [45]. However, after MB treatment, only small Cu hotspots remained suggesting that MB is an effective drug for therapy of WD [45].

Similarly, a significant flush-out of Cu from hepatic tissue in *Atp7b*$^{-/-}$ mice was observed after therapeutic treatment with an adeno-associated virus serotype 8 encoding a therapeutically effective mini version of the human *ATP7B* gene placed under transcriptional control of the liver-specific $\alpha$1-antitrypsin gene promoter [43]. A single treatment with this gene vector was already sufficient to decrease the Cu content to 43.3 ± 3.6 µg/g, while the untreated controls had values of 112.7 ± 13.3 µg/g liver tissue (Figure 7) with coefficient of variation (CV) in measurements that were in the range of 8.3–11.8% [43]. The removal of Cu was further associated with a simultaneous decrease in hepatic Fe, a slight loss in hepatic Zn, reduced urinary copper excretion, increased ceruloplasmin activity, and lower activity of alanine aminotransaminase. All these parameters indicate that this gene therapeutic approach is suitable to abrogate and reverse hepatic alterations occurring during the progression of WD, and further verifies that LA-ICP-MSI is a highly potent technology, strengthening the explanatory power of therapeutic success in such curative studies [43]. In addition, these findings fully confirm the previous atomic absorption spectroscopic measurements and the Timm's sulfide silver stainings showing a reduction in hepatic Cu accumulation after treatment with AAV8-AAT-ATP7B [49].

The beneficial effects to correct cerebral Cu overload by gene reconstitution were also recently confirmed by LA-ICP-MSI in brain sections of treated and untreated *Atp7b*$^{-/-}$ mice [44]. Animals that were sacrificed 14 weeks after therapeutic treatment with AAV-AAT-*co*-miATP7B showed a reduction of mean cerebral Cu content (3.05 ± 0.17 µg/g vs. 3.80 ± 0.2 µg/g) that was most prominently noticeable in the cerebellum, cerebellar white matter, corpus callosum, and 3$^{rd}$ and 4$^{th}$ ventricles when compared to the untreated littermates (Figure 8) [44]. In the brain of respective animals, other elements (Fe, Zn, Mn, Na, Mg, K, Ca, P, Cr, Ni, and Pb) were not affected, suggesting the transgene as an effective mean in specifically directing reduction of cerebral Cu [44].

All these studies demonstrate that LA-ICP-MSI is highly suitable to analyze differences in metal content of wild type and *Atp7b* null mice. Similarly, other methods, including synchrotron-based X-ray fluorescence (SXRF) imaging, were also shown to be beneficial in imaging of copper and other elements in the liver, brain, and intestine of normal and Wilson disease mice. In a previous SXRF study, it was demonstrated that Cu does not continuously accumulate in *Atp7b* null hepatocytes, but reaches a limit at 90–300 fmol [50]. The authors found that the lack of further accumulation was associated with the loss of specific Cu transporters from the plasma membrane and the occurrence of Cu-loaded lymphocytes and extracellular Cu deposits [50]. This technology was further used to demonstrate that *Atp7b* null mice have reduced Cu storage pools in intestine and elevated quantities of iron in enterocytes [17]. It is obvious that multidimensional, high-resolution imaging methods such as SXRF, and lower resolution techniques such as LA-ICP-MSI, are highly interesting for elucidating the metallome of Cu and its changes during disease processes and complex disorders [51].

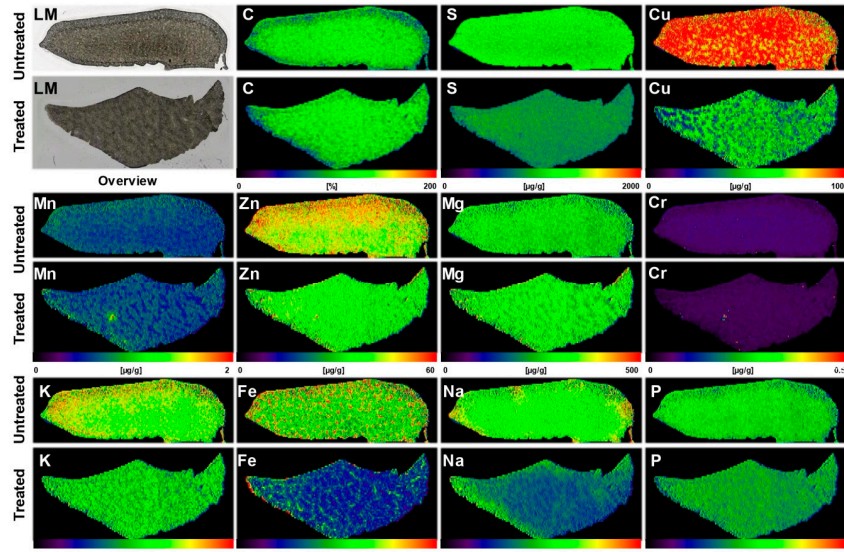

**Figure 7.** Gene therapy in *Atp7b* null mice reverses hepatic Cu overload. The content of carbon (C), sulfur (S), copper (Cu), manganese (Mn), zinc (Zn), magnesium (Mg), chromium (Cr), potassium (K), iron (Fe), sodium (Na), and phosphorus (P) was determined by LA-ICP-MSI in 30-µm-thick liver cryosections of *Atp7b* null mice that received a therapeutic effective AAV virus expressing a therapeutically effective human *ATP7B* gene. Control samples were taken from untreated age-matched *Atp7b* null littermates. For orientation, light microscopic (LM) images of respective cryosections before subjecting to LA-ICP-MSI procedure are depicted. More details of this study are given elsewhere [43,49].

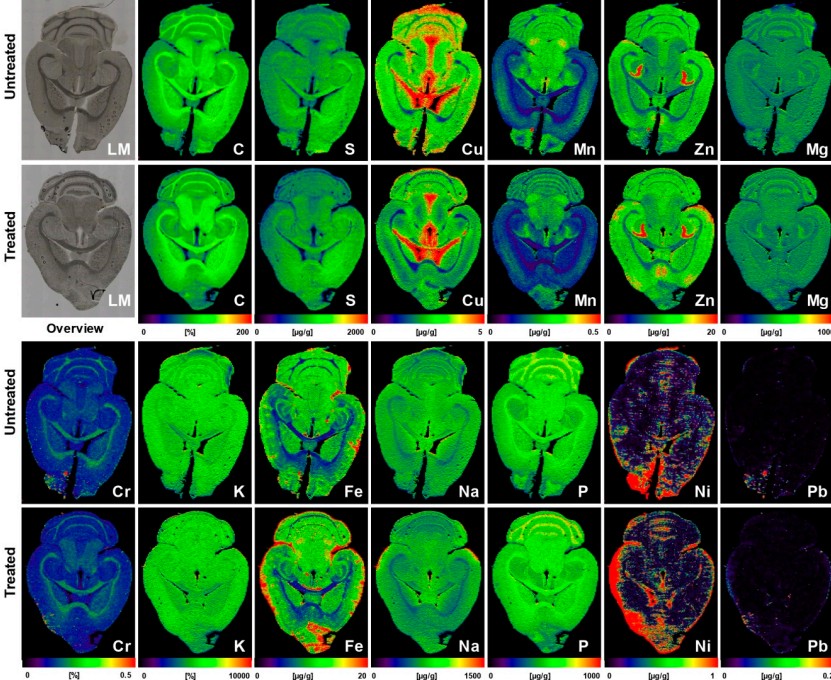

**Figure 8.** Gene therapy in *Atp7b* null mice reverses cerebral Cu overload. The content of carbon (C), sulfur (S), copper (Cu), manganese (Mn), zinc (Zn), magnesium (Mg), chromium (Cr), potassium (K), iron (Fe), sodium (Na), phosphorus (P), nickel (Ni), and lead (Pb) was imaged by LA-ICP-MSI in 30-µm-thick brain cryosections of *Atp7b* null mice that received a therapeutic effective AAV virus expressing a therapeutically effective human *ATP7B* gene. Control samples were taken from untreated age-matched *Atp7b* null littermates. For orientation, light microscopic (LM) images of respective cryosections before subjecting to LA-ICP-MSI procedure are depicted. Details on the applied AAV vector system used in this therapeutic approach are given elsewhere [44,49].

## 6. Translational Aspects of LA-ICP-MSI

All these experimental findings described above demonstrate that gene therapy or Cu chelating drugs are effective in WD to reverse Cu overload. Moreover, LA-ICP-MSI offers great advantages in the diagnosis of Cu overload and its association with other metal alterations. Findings established in experimental models, will also be helpful to manage human WD disease. In WD diagnostic, first studies showed that the Cu overload in liver tissue is well traceable (Figure 9) [39–42].

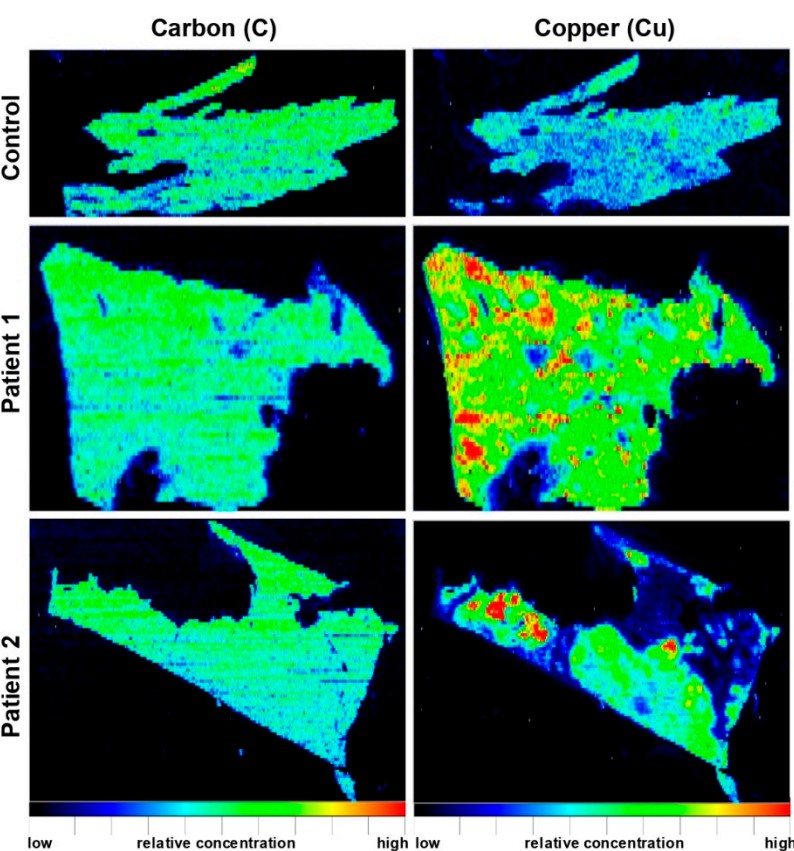

**Figure 9.** LA-ICP-MSI in human Wilson disease livers. Carbon (C) and copper (Cu) were imaged in biopsy samples taken from two Wilson disease patients (patients 1 and 2) and a healthy subject (control). Experimental details of this study are given elsewhere [39].

Therefore, LA-ICP-MSI certainly will become an essential tool in biological research and medical diagnostics. The unique advantages of this technology, including high sensitivity, capability of elemental mapping with a modest lateral resolution, and collection of isotopic information, offer a large variety of applications in life sciences. The results obtained by LA-ICP-MSI discussed in this review for Wilson disease, shows this exemplary.

Human organs are much larger and therefore entire sections of an organ such as the liver cannot be analyzed. In addition, it will not be possible to analyze brain sections from a living human being. Therefore, in some cases LA-ICP-MSI measurements must be combined with other conventional methods or imaging techniques to create diagnostically relevant statements.

In regard to human disease, a relatively high number of recent publications have highlighted some meaningful areas of application for LA-ICP-MSI. We have recently used LA-ICP-MSI to visualize Fe overload in patients suffering from hereditary hemochromatosis [52]. Another previous LA-ICP-MSI investigation from our laboratory has documented the redirection and accumulation of metals within fibrotic human liver tissue [53]. Other laboratories used LA-ICP-MSI or related techniques to analyze the composition of human kidney stones [54], identify inflammatory hotspots associated with elevated

metal quantities in placental tissue from low-birth–weight pregnancy [55], map trace and bulk elements in tumor tissue obtained from an individual previously treated with cisplatin [56], and define contaminations in human hair or teeth resulting from occupational exposure or unfavorable dental fillings [57–59]. Likewise, the first studies analyzed element distribution in human eye lenses, brain tissue, or skin biopsies from patients suffering from renal insufficiency [60–62], optimized radiodosages in treatment of tumors [63], or determined individual metals or metal-containing contrast agents in human blood or serum [64–68]. In addition, there are many inborn errors of metal and metalloid metabolism human trace elements disorders, in which LA-ICP-MSI offers advantages. These are not limited to the extensive repertoire of other Cu-associated disorder, but can lead to alterations in homeostasis of Fe, Zn, Mn, and selenium (Se) [14,69]. Last but not least, LA-ICP-MSI was proposed as a novel complementary tool for forensic pathologists and toxicologists in order to map the presence of metals and other elements in thin tissue of postmortem cases [70].

## 7. Materials and Methods

### 7.1. Images and Pictures

Line drawings in Figures 1, 3 and 5 were done with CorelDRAW Graphics Suite 2018 software from Corel Corporation (Ottawa, Canada).

### 7.2. Animals

Brain samples from untreated wild type and *Atp7b*$^{-/-}$ mice (Figure 6) were kindly provided by Uta Merle and Wolfgang Stremmel (Department of Gastroenterology, University Hospital Heidelberg, Heidelberg, Germany). Additional samples from brain and liver of mice subjected to gene therapy (Figures 7 and 8) were kindly provided by Gloria González-Aseguinolaza and Ruben Hernández-Alcoceba from the University of Navarra, Pamplona, Spain. For this review, new images were generated from data sets published before [38,43,44]. Therefore, no additional animal experimentation was required for this review.

### 7.3. Human Samples

Human liver biopsy samples for histochemical stains (Figure 2) were obtained from the RWTH cBMB biomaterial bank (https://www.cbmb.rwth-aachen.de). Histochemical stains were done by Nikolaus Gassler and Nadine Gaisa (Institute of Pathology, University Hospital Aachen, Aachen, Germany). Additional human liver samples used for LA-ICP-MSI (Figure 9) were provided by Uta Merle and Wolfgang Stremmel (Department of Gastroenterology, University Hospital of Heidelberg, Heidelberg, Germany). The investigations were carried out following the rules of the Declaration of Helsinki of 1975, revised in 2013. Permission to image human liver samples by LA-ICP-MSI was given by the Institutional Ethics Review Board of the Medical Faculty at the RWTH University Hospital Aachen (permit number EK 186/15; date of approval: July 2, 2015). For this review, new images were generated from existing experimental data sets or histochemical stains that were published previously [39,71]. Therefore, no additional human sample materials were needed for the writing of this review.

### 7.4. LA-ICP-MSI Measurements

All measurements were done with a quadrupole-based inductively coupled plasma mass spectrometer (ICP-MS, XSeries 2, Thermo Scientific, Bremen, Germany) coupled to a laser ablation system (NWR 213, New Wave Research, Fremont, CA, USA) in tissue sections of 30 μm thickness. Laser ablation of biological tissue was performed using a focused Nd:YAG laser beam in the scanning mode. The experimental parameters used in LA-ICP-MSI are given in Table 2.

**Table 2.** Typical laser ablation inductively coupled plasma mass spectrometry (LA-ICP-MS) operating parameters for metal imaging in tissue samples.

| Parameter | Experimental Setting |
|---|---|
| ICP mass spectrometer | ICP-QMS (e.g., Thermo XSeries II*) |
| ICP RF power | 1450 W |
| Cooling gas flow rate | 16.0 L·min$^{-1}$ |
| Auxiliary gas flow rate | 0.7 L·min$^{-1}$ |
| Carrier gas flow rate | 1.0 L·min$^{-1}$ |
| Dwell time | 20 ms |
| Extraction lens potential | 3400 V |
| Mass resolution (m/Δm) | 300 |
| Scanning mode | peak hopping |
| Typical analysis time per brain or liver sample (10 mm × 10 mm) | 4 h |
| Laser ablation system | New Wave (NRW213) |
| Wavelength of Nd:YAG** laser | 213 nm |
| Laser fluence | 0.24 J·cm$^{-2}$ |
| Repetition frequency | 20 Hz |
| Laser spot size | 60–80 μm |
| Scan speed | 60 μm·s$^{-1}$ |
| Ablation mode | line scan |

* Several other quadrupole-based ICP-MS instruments equipped with and without a collision/reaction cell from various companies are available from other companies; they differ only marginally in their capabilities. The depicted setting of parameters was optimized for the Thermo XSeries II ICP-MS (Thermo Fisher Scientific, Schwerte, Germany). ** Nd:YAG stands for neodymium-doped yttrium aluminum garnet (Nd:Y$_3$Al$_5$O$_{12}$). These kinds of lasers are used in analysis of elements in the periodic table. It is focused onto the sample surface to produce plasma.

*7.5. Visualization of Data*

All LA-ICP-MSI data were visualized with the modularly constructed in-house software tool Excel Laser Ablation Imaging (ELAI) allowing for the reconstruction of element distribution maps using Microsoft Excel with the aid of Visual Basic for Applications (VBA) [30]. Details about this software tool were published previously [72].

**8. Conclusions**

Metal ions are known to play an important role in many neurodegenerative diseases such as Alzheimer's disease, Parkinsonism, and Amyotrophic Lateral Sclerosis. In WD, genetic defects in the *Atp7b* gene lead to excess Cu accumulation in the liver and later in other organs including the brain. The deposit of Cu is associated with mitochondrial dysfunction, oxidative stress, cell membrane damage, enzyme inhibition, and formation of DNA cross-links. In the liver, the biochemical alterations predispose for liver cirrhosis and hepatocellular carcinoma. Thirty-to-sixty percent of WD patients show a large variety of neurologic and psychiatric symptoms at presentation, including tremor, ataxia, rigidity, dystonia, depressive mood, and many other cognitive and affective disorders. Since hepatic damage and development of neurologic complications is progressive, early detection and treatment of WD patients is crucial in preventing disease progression and organ damage. Presently, examination of liver histology, semiquantitative stainings, and measurement of total Cu in liver biopsy specimen, are generally considered to be the gold standard in diagnosis and monitoring therapeutic success when using chelating agents. However, histochemical Cu stains, chemical titration, and atomic absorption spectrometry measurements are only suitable to visualize or determine mean values of Cu within the tissue. In this regard, LA-ICP-MSI provides spatial information about Cu and other elements in the analyzed tissue sample, thereby allowing a more detailed analysis of individual metals, element networks, or even the complete metallome. Therefore, this technique will be highly helpful to document and visualize the therapeutic efficacy of drugs or other therapy options. It will further provide assistance in experimental studies investigating various aspects of metal overload in tissue or

structures thereof. It is superfluous to stress the fact that LA-ICP-MSI is also meaningful in diagnosis of other metal-associated diseases, in detecting metal poisoning or deficiencies, and documenting forensic matters.

**Author Contributions:** All authors have contributed substantially to this review and agreed to publish this work. R.W.: writing of first draft and preparation of LA-ICP-MS images; S.W.: generation of line drawings; P.K.: preparation of LA-ICP-MSI images.

**Funding:** The laboratory of R.W. is funded by the German Research Foundation (SFB/TRR57, projects P13 and Q3) and the Interdisciplinary Centre for Clinical Research (IZKF) within the Faculty of Medicine at the RWTH Aachen University (Project O3-1). The funders had no role in the design of the study; in the collection, analyses, or interpretation of data; in the writing of the manuscript, or in the decision to publish the results.

**Acknowledgments:** The authors are grateful to Sabine J. Becker, Ricarda Uerlings, Astrid Küppers, and Stephan Küppers, who helped us in LA-ICP-MSI. Special gratitude goes to Wolfgang Stremmel and Uta Merle, who provided samples from untreated *Atp7b* null mice. Gloria González-Aseguinolaza and Ruben Hernández-Alcoceba provided samples from animals that were subjected to *Atp7b* gene therapy. The authors are further grateful to Nikolaus Gassler and Nadine Gaisa for providing histochemical stains of human Wilson disease samples for this review.

**Conflicts of Interest:** The authors declare no conflicts of interest.

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
