# Peer review of "Laser Ablation Inductively Coupled Plasma Spectrometry: Metal Imaging in Experimental and Clinical Wilson Disease"

_inorganics, doi:10.3390/inorganics7040054_

Round 1
Reviewer 1 Report
I consider this paper as very interesting. LA-ICP-MSI seems to be useful method to obtain information about metal distribution in various tissues in animal and human disease. How far it will be useful for clinical practice both the diagnosis and the treatment monitoring I cannot say.
I cannot discuss methodology but parts touching medical parts are well written.
Author Response
Many thanks for taking the time to review our paper. We are happy that you agree that our work is interesting.

Reviewer 2 Report
This is a valuable, well-written and well-presented review that provides a thorough and critical appraisal of the use of LA-ICP-MSI to assess the extent and spatial distribution of copper dysregulation in rodent models and clinical samples of Wilson Disease and to examine the efficacy of potential WD treatments.
Apart from some very minor suggestions regarding the writing (see comments in the attached version of the manuscript) I have no hesitation in recommending this manuscript for publication.

Author Response
Many thanks for taking the time to review our paper. We greatly appreciate the minor errors you found in our draft. We have corrected respective errors in spelling and grammar. In the revised version, all changes are marked in red letters.

Reviewer 3 Report
This is a timely review, that helps the reader to better understand the utility, advantages, and potential applications of LA-ICP-MS methodology in metal related research with a particular focus on Wilson disease. This technology is becoming more accessible and more widely used, therefore, it is likely that the review would elicit interest. The manuscript is clearly written, although careful editing is needed to remove grammatical imperfections and awkward sentences. There are also some other concerns that need to be addressed:
1. The focus of this review is LA-ICP-MS and therefore it is appropriate to focus primarily on a copper imaging work done using this methodology. Nevertheless, a brief comparison to X-ray fluorescence imaging would be highly beneficial to the reader, as advantages and disadvantages of both methodologies could be highlighted This would also allow the authors to acknowledge earlier work by Ralle and colleague on imaging of copper in the liver, brain, and intestine of normal and Wilson disease mice. These references are currently missing
2. The authors must describe how they match the tissue sections to make sure that the observed differences in the copper content in the brain sections reflect copper levels in identical brain region, rather than variability between sections. Such accurate match appears difficult and therefore needs to be discussed.
3. The utility of LA-ICP-MS for research using rodent tissues is clearly demonstrated by excellent illustrations provided by the authors. Human organs are much larger and therefore the entire sections cannot be analyzed. Given variability of copper distribution within the liver (and small and rather random nature of live biopsies) - what is the value of LA-ICP-MS for clinical work? Does it yield any more "actionable" information than the total measurements of copper? This needs to be described more clearly/discussed.
4. As mentioned above, it would be helpful if the text is edited by native English speaker - some sentences and expressions are awkward (see examples below)
1) “…. but might be higher (up to 1:1,130) in 33 communities living socio-culturally isolated [2, 3] –awkward sentence..
2)“A great fraction of excess Cu is not absorbed and defecated, another fraction of this trace element first forms a complex with metallothionein and is then excreted, while the remaining Cu is transported to the liver and incorporated into ceruloplasmin or excreted into the bile” – This is confusing and needs to be better referenced. Higher copper in the diet is associated with higher copper absorption, although effect is non-linear. Please cite the report that show that excess dietary copper first binds to metallothionein (in intestine?) and is then excreted . How/where does this copper get excreted – back into the gut? References or more clear explanations would be helpful
3) Lane 138 "...it is much favorable" - use "much more favorable" or "more favorable"
4) Lane 200 …" raw data files requiring extensively editing.." - use "extensive"
5) Lane 203 ...."Although the plethora of software available used for LA-ICP-MSI data mining is multifarious allowing a lot of representation…” – I do not understand what this means. This entire sentence needs to simplified and probably split in two to better convey the meaning
Author Response
please find our response to all your comments in the attached pdf file.
